# Web Browser Network Based on a BA Model for a Web-Based Virtual World

**Masaki Kohana** [1,*] **, Shinji Sakamoto** [2] **and Shusuke Okamoto** [2]

1    Faculty of Global Informatics, Chuo University, 1-18 Ichigaya-Tamachi, Shinjuku-ku, Tokyo 162-8478, Japan
2    Department of Computer and Information Science, Seikei University, 3-3-1 Kichijoji-Kitamachi,
    Musashino-shi, Tokyo 180-8633, Japan
*    Correspondence: kohana@tamacc.chuo-u.ac.jp

**Abstract:**    Real-time web applications such as a virtual world require considerable computing resources. However, as the number of servers increases, so does the maintenance and financial cost. To share tasks among web browsers, the browsers must share data. Therefore, a network must be constructed among the web browsers. In this paper, we propose the construction of a web browser network based on the Barabasi–Albert model (BA model). We focus on a web-based multiplayer online game that requires higher frequent communication and significant computing resources. We attempt to optimize computing resource utilization for web browsers. We improve upon the method in our previous study, which constructed a network for a web-based virtual world, using only location information. When a new user logged into a world, the web browser connected to two other browsers whose users had a location close to that of the user. The experimental results of that method showed 50% data coverage, which was insufficient to display the game screen because the web browser displays the characters on the virtual world. In this study, we attempt to use the BA model to construct more efficient networks than those in the previous study to increase data coverage. Our new method uses the number of connections of the web browser and location information to calculate the probability of web browser selection. The experimental results show that the data coverage exceeds 90%, indicating significant improvement over the previous method.

**Keywords:** web application; WebRTC; BA model; virtual world

## 1. Introduction

A virtual world is a key feature for creating virtual reality applications and video games. In a virtual world, a user creates and controls their own avatar, and the user moves around the virtual world and interacts with the other users via the avatar. Within this world, the user needs to share the chat message, action, and location of the avatar with the other users. A client-side software displays the virtual world using the shared information.

Virtual world applications involve a server–client architecture. The server stores all of the dynamic information such as the text messages as well as user information such as the appearance of the avatar and the location of the avatars. The client-side software stores static information such as the world images and the sounds, and it retrieves dynamic information from the server. Then, the client-side software displays the game world and the characters.

A traditional type of virtual world divides a game world into many blocks. A server manages blocks, meaning that the server manages the dynamic information regarding the characters on the block. In this type of game world, a user reloads the game world when the user reaches the edge of the block. Communication between servers occurs when the user moves into the other block.

In contrast, open-world virtual worlds are gaining attention. This type of game world appears undivided. The user can move around the world without reloading the map and possesses a higher degree of freedom of movement. However, in an open world, communication between servers occurs frequently because the user can move to neighboring blocks smoothly. Therefore, open-world applications require more frequent communication and considerable computing resources.

In our study, we focus on web-based multiplayer online games owing to the numerous games available and because multiplayer games typically involve some open-world functionality. The web-based game uses web servers to store the information and web browsers as the client-side software. In our previous study, we proposed a system using multiple web servers to construct an open-world web-based game [1].

This system reallocates information that the server should manage to another server to balance the server load. However, the communication traffic becomes a problem. We can balance the amount of communication between the server and the browser. However, an increase in the number of users leads to more frequent communication if we cannot increase the number of servers.

Subsequently, we proposed a method to use web browsers to host information because the number of browsers increases when the number of users increases [2,3]. Furthermore, the web browser displays a part of the virtual world, indicating that the browser requires limited information. This could possibly reduce the number of communications. We try sharing the information using a network of web browsers. Our system constructs a web browser network by using web real-time communication (WebRTC) [4] to provide features to communicate among web browsers without web servers. This network achieves approximately 50% of data coverage, which is insufficient for the virtual world. The web browser cannot display the complete game world. For example, the browser cannot display the other users even though the user exists in the game world. This situation inhibits the fairness and consistency of the game. Therefore, we must achieve 100% data coverage.

To increase data coverage, we need to construct a more efficient network. In this paper, we use the Barabási–Albert model (BA model), which is an algorithm that constructs a scale-free network. In an online game, the user logs into the virtual world. Therefore, the new user is the new node in the network. The number of nodes gradually increases. We consider that this characteristic is similar to the BA model. Therefore, we consider that the BA model is familiar with our system.

The remainder of the paper is organized as follows. Section 2 introduces some research related to our study. Section 3 describes an overview of our method to construct a web browser network. Section 4 shows the experimental results, and Section 5 provides conclusions.

## 2. Literature Survey

This section introduces some studies related to our work.

Xhafa et al. investigated several application models for mobile computing [5]. In their paper, they focused on collaborative work with geographically distributed members. In this paper, we introduce the peer-to-peer model using WebRTC and a central server model and discuss some problems that arise in application migration such as data synchronization and consistency.

WebRTC makes a web browser that connects to the other web browser, which means that the web browser can construct peer-to-peer networks. Vogt et al. discusses leveraging WebRTC to distribute content for web browsers [6].

WebRTC can also be used for media streaming. Jukka et al. discussed the use of WebRTC for peer-to-peer(P2P) media streaming [7]. This is an essential topic for video-on-demand.

Ito et al. also proposed a way to stream video content for a small classroom [8]. This system constructs a tree network using clients of the students. The computer of the teacher becomes the root node and delivers the content.

Petrangeli et al. described an issue concerning WebRTC [9]. The issue is that the sending peer in a WebRTC session requires an independent stream for each receiving peer. To address this, the authors propose a framework. This framework uses a conference controller that is a centralized node

based on the selective forwarding unit (SFU). This controller provides the best stream to the receiving peer based on the bandwidth condition. Moreover, the sending peer transmits to multiple receiving peers simultaneously.

Seo et al. mentions the P2P-assisted DASH technology based on WebRTC to reduce the content delivery network (CDN) cost [10]. The author proposes an algorithm that selects a peer based on the history of the transport. This paper uses the transport history.

These studies on WebRTC focus on video streaming or CDN. They mention the issue of network congestion, transportation path, and selecting a peer. In our study, we attempt to solve these problems by using a small network based on location information in the virtual world.

Another key technology of our study is the scale-free network. We use this to model several networks.

Kuhn et al. investigated distributed computation in a dynamic network [11]. In this paper, the network topology changes. Moreover, they consider the worst-case model. The considered model is similar to our assumption that the Web browsers construct networks dynamically.

Kang proposed a fog computing model that is safe, stable, and efficient [12]. This paper modeled the evolution process of fog computing based on the BA and ER models. Moreover, the author obtained a network model with two network characteristics.

Lin et al. demonstrated the importance of transmission costs for a network routing strategy [13]. The costs include route, distance, and traffic costs. This paper evaluated theoretical and real networks. Moreover, this paper shows the importance of transmission costs to improve the effectiveness of the network routing strategy.

Zhang et al. focused on social networks [14]. Their paper proposed a network model that simulates human daily workspace and multi-residential areas.

In our study, our system constructs a web browser network based on location information in a virtual world. Moreover, in this network, the web browser transmits information to share it with adjacent browsers. Therefore, our system uses the BA model, which is a scale-free network, and the location information in a virtual world.

## 3. System Overview

This section describes an overview of our method to construct a web browser network. Our system is a web-based multiplayer online game with an open-world model. For this game type, a user creates an avatar character in a virtual world. All users share the same virtual world, and a user interacts with the other users via the avatar. For the interaction, the user needs to share their own information such as text messages, avatar actions, and location of the avatar with other users. Traditional web-based online games share the information via a web server. However, in our system, the user sends the information to the other users without a web server because our system constructs the web browser network. In this network, a web browser connects to other browsers directly. Therefore, our system avoids communication congestion in the web server.

Figure 1 shows a screenshot of our virtual world application. There are two avatars, an elephant and a frog. The user who controls the elephant locates their own avatar at the center of the game screen. The other user controls the frog. The blue spheres are non-player characters (NPCs). A server-side program controls the NPCs. This screenshot is a part of the virtual world, which means that the web browser displays a part of the world. Therefore, the web browser does not require all of the information. It only needs the information of the characters that are near their own avatar.

Our system needs to construct a web browser network. Moreover, the web browser should collect the necessary information using the network. To construct this network, we use two technologies—WebRTC and the Barabási–Albert model (BA model).

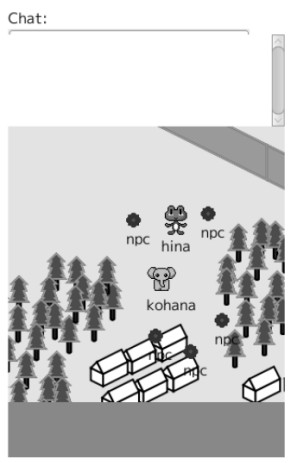

**Figure 1.** Screenshot.

### 3.1. WebRTC

WebRTC is a communication protocol among web browsers. W3C defines the Application Programming Interface(API) specification [4], and Internet Engineering Task Force(IETF) standardizes the protocol [15]. Using WebRTC, web browsers can construct a peer-to-peer network. On this network, the communication does not use a web server as a communication path. Therefore, web browsers can communicate with the other browsers without any backend server. The WebRTC has two channels, the media stream for multimedia data transfer and the data channel for other binary data transfers. In our study, we construct a web browser network using the WebRTC. The client-side software transfers the information via the data channel of WebRTC, while the client exchanges the information via a web server in traditional web-based computer games. In our system, the web server provides static information such as the character images, world images, and the sounds. Each client-side software program manages dynamic information, which is the position of its own avatar. Subsequently, the client sends information about the avatar and the text message via the data channel of WebRTC. Using this method, the client shares the dynamic information without any communication with the web server.

### 3.2. Barabási–Albert Model (BA Model)

The Barabási–Albert model (BA model) is an algorithm that constructs a scale-free network. This algorithm stands on the growth and the preferential attachment mechanism.

In this algorithm, the initial state comprises a complete graph with $m_0$ nodes. When a new node joins the network, the node creates $m$ connections with the existing nodes based on the preferential attachment mechanism. If the number of nodes is $n$ and the frequency $k_i$ of the existing node $v_i$, the probability that the new node connects to the node $v_i$ is given by the following expression (1):

$$p_i = \frac{k_i}{\sum_{j=0}^{n} k_j}. \tag{1}$$

Figure 2 shows an example of network construction based on the BA model. In this figure, the value $m_0$ is 3, and the number $m$ is 2. In this case, the value $k_i$ is the number of connections. The three nodes construct a complete graph. When node D joins the network, the probability of each node is 2/3. Therefore, node D choose two nodes and connects to these nodes. After that, node E joins the network and also chooses two nodes with a higher probability. However, the new node might connect to the lower probability node. For example, node F connects to node B.

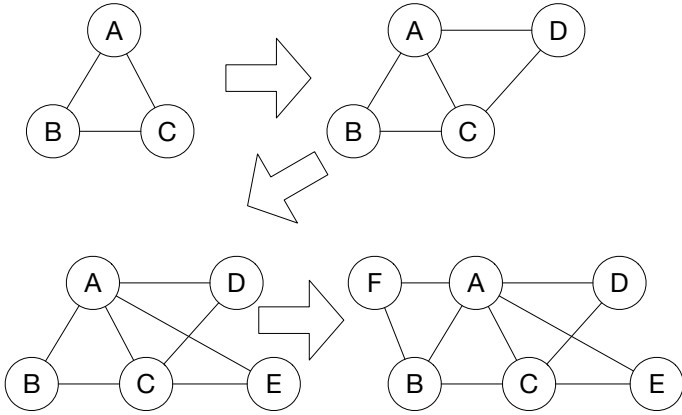

**Figure 2.** Example of the Barabasi–Albert(BA) model.

### 3.3. System Configuration

Figure 3 shows the construction of the virtual world. Our setting divides the world into small blocks. Our system manages the blocks in units called as chunks. Each chunk comprises 9 (4 × 4) blocks. The entire world comprises 18 (6 × 3) chunks. An avatar exists in only one block.

Figure 4 shows an example of a view range of a client. There are two avatars, an elephant and a frog. A web browser needs information about the characters in a view range to display a game screen. The browser collects information of a chunk. The gray regions denote the view range of the elephant and the frog. For example, the elephant exists in a block that is at the center of the chunk. The browser of the elephant loads information about the characters that are included in the gray chunks. In contrast, the frog exists in a block that is at the top-right side of the chunk. However, the browser loads information in a chunk. Therefore, the browser of the frog also loads information about the characters in nine gray chunks.

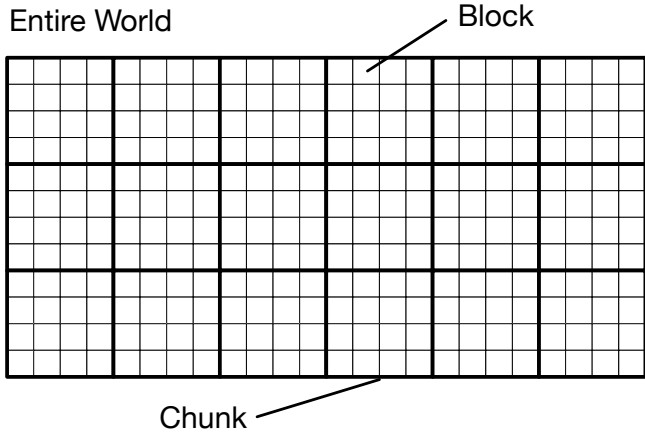

**Figure 3.** Dividing world.

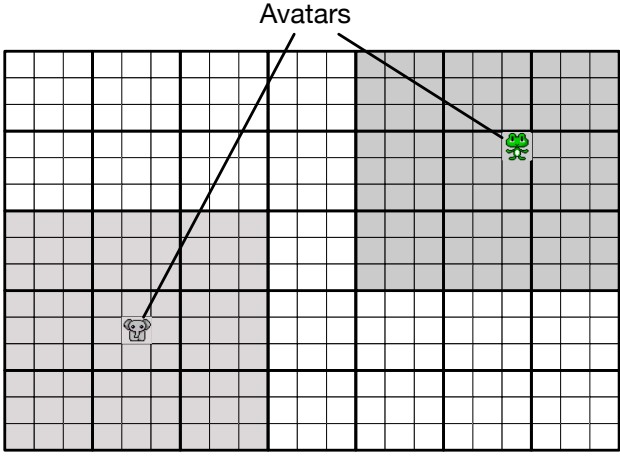

**Figure 4.** Display range.

*3.4. Network Construction*

Our system has a web server and many web browsers. The server manages the network topology, and the browsers are client-side software that stores information regarding characters and displays a game screen.

Figure 5 shows the construction of a browser network. The server selects two browsers that the new browser should connect to. Therefore, browser A in this figure sends a request to the server. Then, the server chooses two other browsers based on the BA model. The server chooses browsers B and C and relays this information to browser A. Afterwards, browser A sends a connection request to browsers B and C.

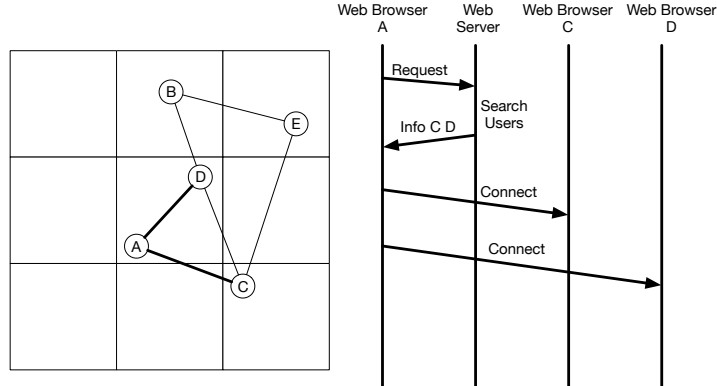

**Figure 5.** Network construction.

Our system limits the number of connections to two when the new user logs into the virtual world. We understand that the system can reach 100% data coverage if all clients connect to all other clients. In this situation, the number of connections for a client is the same as the number of users. However, the web browser has a limited number of network connections because the number of TCP sockets on the operating system is limited. Therefore, if the number of users exceeds the number of connections, the browser cannot connect to all the other users. On the other hand, the parallelism for the communication depends on the number of cores on the client computer. The number of communication processes is the same as the number of the other users. However, the number of processes that the client can run simultaneously is the same as the number of cores. The communication

process can be separated from the user interface process on the web browser. Therefore, the connection process does not affect the operability of the user interface. However, the data translation gets delayed because a communication process waits for the other communication processes. Consequently, the client does not receive information on all of the other users. Therefore, the client does not display the complete game world.

In our previous study, our system chose two existing clients when the new user logs into the virtual world. One client exists in the same chunk as the new user. The second client is selected from the chunks adjacent to the chunk where the new user exists. In this manner, we could create a network and share the information. However, the network size tends to be small. The experimental result shows that the data coverage is about 50%, which is inadequate. The result indicates that the web browser cannot display the complete virtual world with the avatars of other users. Therefore, in this paper, we use an algorithm to create a scale-free network.

The server chooses the target clients based on the BA model. In our system, to calculate the probability, we use the number of connections and location information. The location information is the position of the avatar in the virtual world. The web browser requires the information of the chunks that are included in the view range. Therefore, the browser should connect to the nearby browsers. Expression (2) indicates probability calculation. The value $d_{ij}$ indicates the distance between avatar $i$ and avatar $j$. The value $ws$ indicates the world size. In our system, the world size is the number of blocks at the longest-side edge. If the world comprises $1500 \times 1000$ blocks, the world size is 1500:

$$p_i = (1 - \frac{d_{ij}}{ws}) + \frac{k_i}{\sum_{j=0}^{n} k_j}. \tag{2}$$

Figure 6 shows an example of network construction using our method. In this figure, there are six white nodes and one gray node. The gray node indicates the new node, whereas the white nodes indicate existing nodes. Each white node has the probability that is the summation of the value based on the distance and the value based on the number of connections. The left-side value is the probability of the distance, and the right-side value is that of the number of connections. Using the traditional BA model, the new gray node connects to the nodes that have a higher number of connections. However, our method considers the distance between the new node and the existing node because the new node needs the information included in the view range. Therefore, the new node connects to nodes C and F even though node A has a higher probability based on the number of connections.

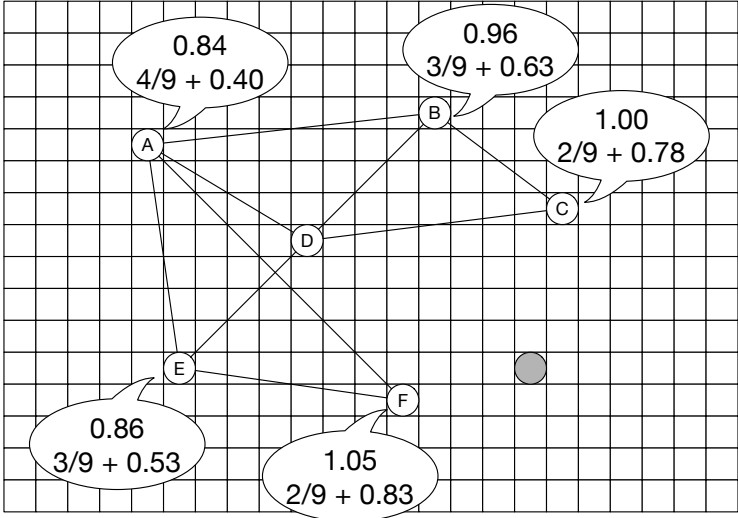

**Figure 6.** Example of the BA model with location.

### 3.5. Data Transfer

In our system, each web browser sends information about the characters using WebRTC. However, a web browser needs the information included in the view range, which means that the browser does not need the information about all characters. Therefore, our system controls the information range based on the view range of the avatar.

Figure 7 shows an example of the controlling information range. As the first step, avatar A sends information to avatars B and C. Avatar B sends the received information to avatar E because avatar B receives the information from avatar A. Avatar C also sends the information to avatar D. After that, avatar E sends the information to avatars D and F, and avatar D also sends the information to avatars E and F. Avatar F exists in the out-of-view range of avatar A, and avatars G and H also exist in the out-of-view range. In this situation, avatar F does not transfer the information to G and H.

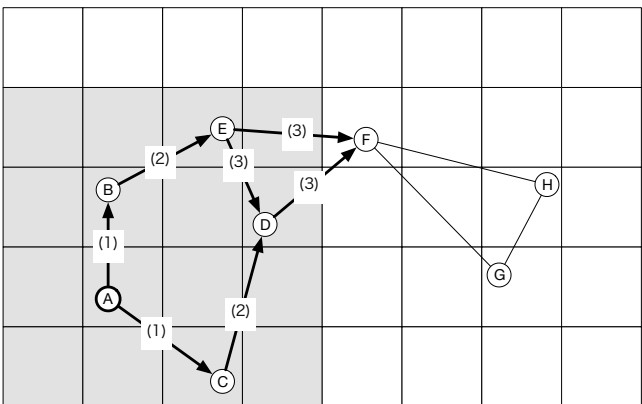

**Figure 7.** Data transfer.

## 4. Experimental Results

This section illustrates the result of our simulation experiment. In the experiment, we measured the data cover rate and the number of connections. Expression (3) shows the data cover rate:

$$CoverRate = \frac{Nrec}{Nreq}. \tag{3}$$

*Nrec* indicates the amount of information received by other users. *Nreq* indicates the amount of information required to display the game world. In other words, *Nreq* indicates the number of users in the view range of the user. If a user needs five user data elements to display the game screen and receives three user data elements, the data cover rate is $3/5 (60\%)$. In our simulation-based experiment, the size of the virtual world is $1500 \times 1500$. The chunk size is $15 \times 15$, indicating that the world comprises $100 \times 100$ (10,000) chunks. The view range is 10 chunks in one direction. If the user exists at the position $(25, 25)$, the top-left side of the view range is $(15, 15)$, and the bottom-right side of the view range is $(35, 35)$. Therefore, the view range includes 441 chunks. The configuration in our experiment refers to the default configuration of Minecraft [16].

Figure 8 shows the data cover rate using our previous model. Figure 9 shows the result using the BA model. The *x*-axis indicates the number of users, whereas the *y*-axis indicates the data cover rate. The range of the *y*-axis on Figure 8 is 0–1, whereas that on Figure 9 is 0–0.1, which is $\frac{1}{50}$ of Figure 8 because the data cover rate is less than 10% using the BA model, implying that the user cannot share the most information. We cannot accept this result because the web browser cannot display most of the game world with this data cover rate. The cause of this result is that our data transfer method is not familiar with the network based on the BA model. Our data transfer method considers the view range.

If an avatar receives the information but exists in the out-of-view range, the avatar does not transfer the information. However, the BA model considers the number of connections as the probability for the network construction. Therefore, the node may transfer the data to another node that exists in the out-of-view range.

Figure 10 shows the data cover rate using the BA model and the location information. This method considers the number of connections and location information. The data cover rate achieves more than 90%. Using the BA model and location information, a user can collect the most information. However, we want to achieve 100% data cover rate to display the complete game world. We believe that this cannot be achieved with the current data transfer method. Thus, in future work, we must improve the method of the data transfer.

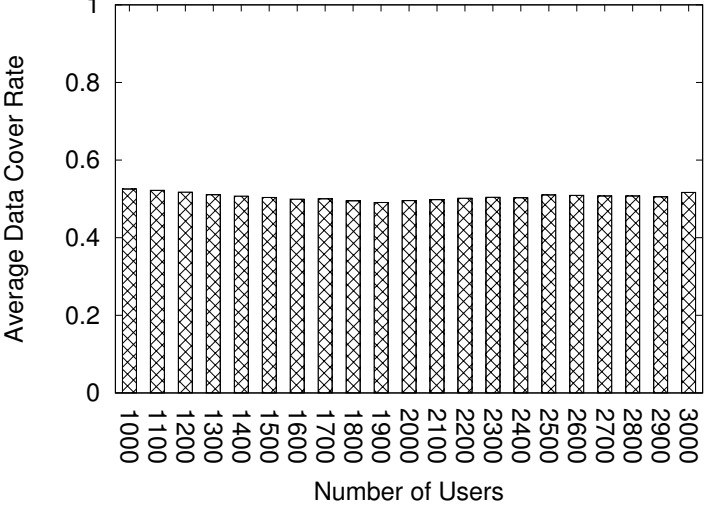

**Figure 8.** Data coverage with the previous model.

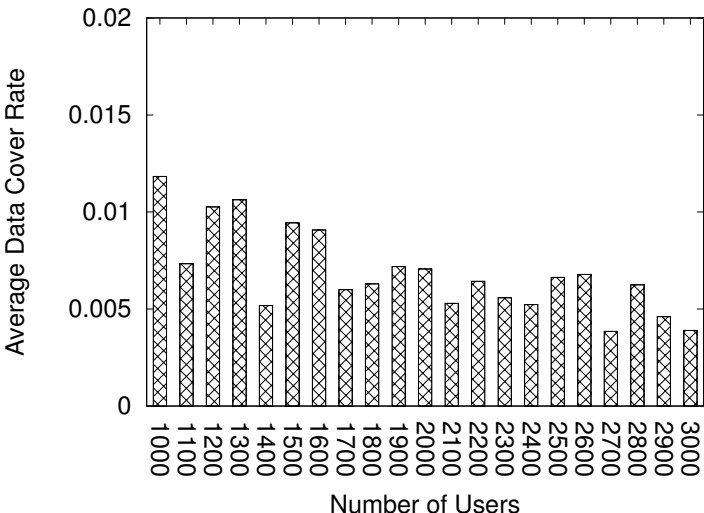

**Figure 9.** Data coverage with the BA model.

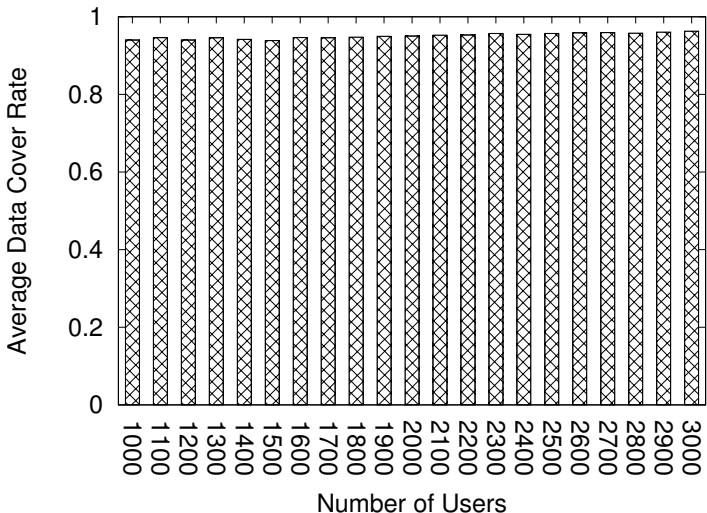

**Figure 10.** Data coverage with BA model and location.

## 5. Conclusions

In this paper, we propose a method to construct a web browser network using the BA model. In our proposed model, a new user joins our system, the web server selects the target users, and the user connects to the targets. In this time, our system uses the BA model to choose the targets. However, our system considers location information to transfer data, whereas the traditional BA model does not. Therefore, the data cover rate is less than 10%. Our proposed method calculates the probability using the distance information to modify the BA model. Our system considers the location information and the traditional BA model. As a result, the achieved data cover rate exceeds 90%. However, as our system does not consider the users' movements and them exiting the world, our system needs a reconstruction of the network. This will be addressed in a future study.

**Author Contributions:** conceptualization, M.K., S.S and S.O.; methodology, M.K., S.S., S.O.; software, M.K.; validation, M.K., ; formal analysis, M.K.; investigation, M.K., S.S.; resources, M.K.; data curation, M.K.; writing—original draft preparation, M.K.; writing—review and editing, M.K.; visualization, M.K.; supervision, S.O.; project administration, M.K.;

**Funding:** This research received no external funding

**Acknowledgments:** We would like to thank Editage (www.editage.com) for English language editing.

**Conflicts of Interest:** The authors declare no conflict of interest.

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
