# Peer review of "Web Browser Network Based on a BA Model for a Web-Based Virtual World"

_futureinternet, doi:10.3390/fi11070147_

Reviewer 1 Report

The authors propose a way to create a scale free network of a virtual game world using web browsers that connect to each other using an adaptation of the BA model that uses the location information. The idea is that by doing this they are able to decrease the number of servers while at the same time maintain the data coverage.

The Abstract should also refer that the paper is about the computer game virtual world.

Chapter 2. Literature Survey is very limited, does not even refer any research using the Barabási–Albert model (that is presented in chapter 3), and does not discuss how the proposed approach relates with the works cited.

In Chapter 3 there is little information on how the WebRTC is used in the Web browsers' communication using the Data Channel.

In Chapter 3.4, what is the location information of the browsers? Is the geographic location of the browsers in the real world or the location of the avatar in the virtual world? It seems that it is from the virtual world, however this is not crystal clear.

Regarding the use of the distance with the BA model: there is no dij in equation 2. What is the meaning of world size? The number of blocks or chuncks, or something else? Is there a limit on the number of connections a web browser is able to accept? What happens when this limit is reached? How about the different Internet connection speed of each browser? Does this affect the browser network overall performance? How does this overhead of connections affect the user speed of operation? How does it affect the user computer?

In chapter 4 the authors say "The data cover rate shows the number of necessary data received by a user." and I do not agree with the definition. How can a rate, which should be a percentage, be the number of data?

In line 144, figure 1 should be replaced by Figure 9

The scale of Figure 9 does not allow to see the actual values of the bars.

The paper has a huge number of English problems that must be fixed before publishing. The number is so large that I cannot enumerate them. This is a very big problem that must be taken care of before publishing, because how it is, for me it is unacceptable for publication. I advise the authors to use professional help in this matter.

Author Response

Dear Editors:

We wish to re-submit the manuscript titled "Web Browser Network based on BA Model for Web-based Virtual World."

The manuscript ID is futureinternet-527543.

We thank you and the reviewers for your thoughtful suggestions and insights.

The manuscript has benefited from these insightful suggestions. 

I look forward to working with you and the reviewers to move this manuscript closer to publication in the Future Internet.

The manuscript has been rechecked and the necessary changes have been made in accordance with the reviewers’ suggestions. The responses to all comments have been prepared and attached herewith/given below. 

Thank you for your consideration. I look forward to hearing from you.

We highlight the changed part of our manuscript.

And we list the points of the change below with the page number and the line number.

First of all, our manuscript undergoes English editing by a native English speaker.

Reviewer 1 Abstract

pp. 1, line 4-6:

We add the target of our study.

Reviewr 2 Comment 1.

We rephrase the introduction section.

Reviewer 2 Comment 4

pp. 1, line 8-13

pp. 2, line 39-47:

pp. 6, line 180-186:

We add the description about our previous work.

pp. 1, line 16:

We add the achievement of our study.

Reviewer 2 Comment 3

pp. 1, line 24-38:

We describe the architecture of multiplayer online games.

Reviewr 2 Comment 2

pp. 2, line 56-59:

We describe a reason why we use BA model.

Reviewer 1 Chapter 2

Reviewer 2 Comment 2

pp. 2, line 77-85:

pp. 3, line 94-102:

We add some related works about webRTC and scale-free network.

Reviewer 1 Chapter 2

Reviewer 2 Comment 2

pp. 3, line 86-90:

pp. 3, line 103-106:

We describe the relationship between the related works and our study.

Reviewr 2 Comment 3

pp. 3, line 108-122:

We describe our multiplayer onlien game system.

And we describe the relationship between Fig. 2 and our system.

Reviewer 1 Chapter 3

pp. 4, line 134-139:

We describe how to use webRTC.

Reviewer 1 Chapter 3.4

pp. 6, line 174-179:

We add the description about the limitation of the number of connection on a web browser.

However, we do not consider the Internet speed for each web browser currently.

And we do not measure the affect for the Internet performance.

Reviewer 1 Chapter 3.4

pp. 6, line 188-190:

We describe what is the location information.

Reviewer 1 Chapter 3.4

expression (2):

We fix the dij.

Reviewer 1 Chapter 3.4

pp. 7, line 192-193:

We add the description about the world size.

Reviewr 1 Chapter 4

pp. 8, line 217-219:

expression (3):

We define and describe what is the data cover rate.

Reviewr 1 Figure 9

pp. 8, line 226:

We fix the figure number.

Reviewr 1 Figure 9

pp. 9, Figure 9:

We modify the scale of the graph.

The following part is the original comments from the reviewers.

Reviewer1:

The authors propose a way to create a scale free network of a virtual game world using web browsers that connect to each other using an adaptation of the BA model that uses the location information. The idea is that by doing this they are able to decrease the number of servers while at the same time maintain the data coverage.

The Abstract should also refer that the paper is about the computer game virtual world.

Chapter 2. Literature Survey is very limited, does not even refer any research using the BarabsiAlbert model (that is presented in chapter 3), and does not discuss how the proposed approach relates with the works cited.

In Chapter 3 there is little information on how the WebRTC is used in the Web browsers’ communication using the Data Channel.

In Chapter 3.4, what is the location information of the browsers? Is the geographic location of the browsers in the real world or the location of the avatar in the virtual world? It seems that it is from the virtual world, however this is not crystal clear.

Regarding the use of the distance with the BA model: there is no dij in equation 2. 

What is the meaning of world size? The number of blocks or chuncks, or something else? Is there a limit on the number of connections a web browser is able to accept? What happens when this limit is reached? How about the different Internet connection speed of each browser? Does this affect the browser network overall performance? How does this overhead of connections affect the user speed of operation? How does it affect the user computer?

In chapter 4 the authors say ”The data cover rate shows the number of necessary data received by a user.” and I do not agree with the definition. How can a rate, which should be a percentage, be the number of data?

In line 144, figure 1 should be replaced by Figure 9

The scale of Figure 9 does not allow to see the actual values of the bars.

The paper has a huge number of English problems that must be fixed before publishing. The

number is so large that I cannot enumerate them. This is a very big problem that must be taken care of before publishing, because how it is, for me it is unacceptable for publication. I advise the authors to use professional help in this matter.

Reviewer2:

This paper proposed a WebRTC-based peer-to-peer network among web browsers. They use BA model to decide which peers should be connected to get the context data coverage. Their method calculate the probability using the distance as a new impact factor in BA model to change its decisions on the target browsers. The work is very interesting and the result is very useful. Several places for improvements:

1. The introduction part is confusing. It seems the target scenario of this network is online video game. Readers will struggle to understand here. The whole section should be rephrased. Make sure it goes smoothly with the order from the background, reason why you create this WebRTC network, general introduction of the existing work and your work, and the achievement you eventually got.

2. The literature review is not enough. There are many recent researches on WebRTC browser network and the state-of-the-art should be fully reviewed.

3. System overview should not be some simple introduction of already existing technology, also introduce the original work you have created upon them. The screenshot of the virtual world application is very shy and have no idea about its purpose. It would be better if the paper could explain the game or the virtual world in details.

4. Experimental result:“Figure 8 shows the data cover rate using our previous model”What previous model? Besides your old model, are there any existing widely used models (mentioned in literature review) that can be compared with?

Reviewer 2 Report

This paper proposed a WebRTC-based peer-to-peer network among web browsers. They use BA model to decide which peers should be connected to get the context data coverage. Their method calculate the probability using the distance as a new impact factor in BA model to change its decisions on the target browsers. The work is very interesting and the result is very useful. Several places for improvements:

1. The introduction part is confusing. It seems the target scenario of this network is online video game. Readers will struggle to understand here. The whole section should be rephrased. Make sure it goes smoothly with the order from the background, reason why you create this WebRTC network, general introduction of the existing work and your work, and the achievement you eventually got.  

2. The literature review is not enough. There are many recent researches on WebRTC browser network and the state-of-the-art should be fully reviewed.

3. System overview should not be some simple introduction of already existing technology, also introduce the original work you have created upon them. The screenshot of the virtual world application is very shy and have no idea about its purpose. It would be better if the paper could explain the game or the virtual world in details.

4. Experimental result: “Figure 8 shows the data cover rate using our previous model” What previous model? Besides your old model, are there any existing widely used models (mentioned in literature review) that can be compared with?

Author Response

Dear Editors:

We wish to re-submit the manuscript titled "Web Browser Network based on BA Model for Web-based Virtual World."

The manuscript ID is futureinternet-527543.

We thank you and the reviewers for your thoughtful suggestions and insights.

The manuscript has benefited from these insightful suggestions. 

I look forward to working with you and the reviewers to move this manuscript closer to publication in the Future Internet.

The manuscript has been rechecked and the necessary changes have been made in accordance with the reviewers’ suggestions. The responses to all comments have been prepared and attached herewith/given below. 

Thank you for your consideration. I look forward to hearing from you.

We highlight the changed part of our manuscript.

And we list the points of the change below with the page number and the line number.

First of all, our manuscript undergoes English editing by a native English speaker.

Reviewer 1 Abstract

pp. 1, line 4-6:

We add the target of our study.

Reviewr 2 Comment 1.

We rephrase the introduction section.

Reviewer 2 Comment 4

pp. 1, line 8-13

pp. 2, line 39-47:

pp. 6, line 180-186:

We add the description about our previous work.

pp. 1, line 16:

We add the achievement of our study.

Reviewer 2 Comment 3

pp. 1, line 24-38:

We describe the architecture of multiplayer online games.

Reviewr 2 Comment 2

pp. 2, line 56-59:

We describe a reason why we use BA model.

Reviewer 1 Chapter 2

Reviewer 2 Comment 2

pp. 2, line 77-85:

pp. 3, line 94-102:

We add some related works about webRTC and scale-free network.

Reviewer 1 Chapter 2

Reviewer 2 Comment 2

pp. 3, line 86-90:

pp. 3, line 103-106:

We describe the relationship between the related works and our study.

Reviewr 2 Comment 3

pp. 3, line 108-122:

We describe our multiplayer onlien game system.

And we describe the relationship between Fig. 2 and our system.

Reviewer 1 Chapter 3

pp. 4, line 134-139:

We describe how to use webRTC.

Reviewer 1 Chapter 3.4

pp. 6, line 174-179:

We add the description about the limitation of the number of connection on a web browser.

However, we do not consider the Internet speed for each web browser currently.

And we do not measure the affect for the Internet performance.

Reviewer 1 Chapter 3.4

pp. 6, line 188-190:

We describe what is the location information.

Reviewer 1 Chapter 3.4

expression (2):

We fix the dij.

Reviewer 1 Chapter 3.4

pp. 7, line 192-193:

We add the description about the world size.

Reviewr 1 Chapter 4

pp. 8, line 217-219:

expression (3):

We define and describe what is the data cover rate.

Reviewr 1 Figure 9

pp. 8, line 226:

We fix the figure number.

Reviewr 1 Figure 9

pp. 9, Figure 9:

We modify the scale of the graph.

The following part is the original comments from the reviewers.

Reviewer1:

The authors propose a way to create a scale free network of a virtual game world using web browsers that connect to each other using an adaptation of the BA model that uses the location information. The idea is that by doing this they are able to decrease the number of servers while at the same time maintain the data coverage.

The Abstract should also refer that the paper is about the computer game virtual world.

Chapter 2. Literature Survey is very limited, does not even refer any research using the BarabsiAlbert model (that is presented in chapter 3), and does not discuss how the proposed approach relates with the works cited.

In Chapter 3 there is little information on how the WebRTC is used in the Web browsers’ communication using the Data Channel.

In Chapter 3.4, what is the location information of the browsers? Is the geographic location of the browsers in the real world or the location of the avatar in the virtual world? It seems that it is from the virtual world, however this is not crystal clear.

Regarding the use of the distance with the BA model: there is no dij in equation 2. 

What is the meaning of world size? The number of blocks or chuncks, or something else? Is there a limit on the number of connections a web browser is able to accept? What happens when this limit is reached? How about the different Internet connection speed of each browser? Does this affect the browser network overall performance? How does this overhead of connections affect the user speed of operation? How does it affect the user computer?

In chapter 4 the authors say ”The data cover rate shows the number of necessary data received by a user.” and I do not agree with the definition. How can a rate, which should be a percentage, be the number of data?

In line 144, figure 1 should be replaced by Figure 9

The scale of Figure 9 does not allow to see the actual values of the bars.

The paper has a huge number of English problems that must be fixed before publishing. The

number is so large that I cannot enumerate them. This is a very big problem that must be taken care of before publishing, because how it is, for me it is unacceptable for publication. I advise the authors to use professional help in this matter.

Reviewer2:

This paper proposed a WebRTC-based peer-to-peer network among web browsers. They use BA model to decide which peers should be connected to get the context data coverage. Their method calculate the probability using the distance as a new impact factor in BA model to change its decisions on the target browsers. The work is very interesting and the result is very useful. Several places for improvements:

1. The introduction part is confusing. It seems the target scenario of this network is online video game. Readers will struggle to understand here. The whole section should be rephrased. Make sure it goes smoothly with the order from the background, reason why you create this WebRTC network, general introduction of the existing work and your work, and the achievement you eventually got.

2. The literature review is not enough. There are many recent researches on WebRTC browser network and the state-of-the-art should be fully reviewed.

3. System overview should not be some simple introduction of already existing technology, also introduce the original work you have created upon them. The screenshot of the virtual world application is very shy and have no idea about its purpose. It would be better if the paper could explain the game or the virtual world in details.

4. Experimental result:“Figure 8 shows the data cover rate using our previous model”What previous model? Besides your old model, are there any existing widely used models (mentioned in literature review) that can be compared with?

Round  2

Reviewer 1 Report

I acknoledge that the authors made an effort to improve this version of the paper.

The authors propose a way to create a scale free network of a virtual game world using web browsers that connect to each other using an adaptation of the BA model that uses the location information. The idea is that by doing this they are able to decrease the number of servers while at the same time maintain the data coverage.

In the Abstract there is "...In this method...result show 50% data coverage" and this "The experimetn shows the data coverage is more than 90%".

In Chapter 3.4, the authors do not discuss how serious is the fact that a browser is not capable of displaying 100% of the world? How do you solve this? Do you need to access a server in those situations? 

Regarding the previous questions that were not considered in this updated version of the paper, the authors may discuss what are the threats for validity of their proposal. That's what those questions were mostly about, but there are many others that could be addressed. Here are the previiuos questions: Is there a limit on the number of connections a web browser is able to accept? What happens when this limit is reached? How about the different Internet connection speed of each browser? Does this affect the browser network overall performance? How does this overhead of connections affect the user speed of operation? How does it affect the user computer?

The scale of Figure 9 should be further improved in order to better see the actual values of the bars, in in the text referring to it it should be pointed out the scale used and why.

The paper has a huge number of English problems that must be fixed before publishing. The number is so large that I cannot enumerate them. This is a very big problem that must be taken care of before publishing, because how it is, for me it is unacceptable for publication. I advise the authors to use professional help in this matter. For example, almost all sentences of the Abstract have mistakes, like this one: "However, resourices on the web servers, the number of servers increases, which leads to an increase in maintenance and monetary cost."

Author Response

Dear Editors:

We wish to re-submit the manuscript titled "Web Browser Network based on BA Model for Web-based Virtual World."
The manuscript ID is futureinternet-527543.

We thank you and the reviewers for your thoughtful suggestions and insights. The manuscript has benefited from these insightful suggestions.
I look forward to working with you and the reviewers to move this manuscript closer to publication in the Future Internet.

The manuscript has been rechecked and the necessary changes have been made in accordance with the reviewersʼ suggestions. The responses to all comments have been prepared and attached herewith/given below.

Thank you for your consideration. I look forward to hearing from you.

Our manuscript is edited, and the abstract is rearranged by the English native speaker with Editage.
I add the certification of the English Editing.

We highlight the revised parts of our manuscript.

pp.2, line 53-56:
pp.9, line 250-252:

We describe how affect the data coverage to the displaying game world based on the following reviewerʼs comment.

In the Abstract there is "...In this method...result show 50% data coverage" and this "The experiment shows the data coverage is more than 90%".
In Chapter 3.4, the authors do not discuss how serious is the fact that a browser is not capable of displaying 100% of the world? How do you solve

this? Do you need to access a server in those situations?

pp.6, line 176-189:
We add the description about the overhead and performance if the number of connections on web browser reaches the limitation according to the following comment.

Regarding the previous questions that were not considered in this updated version of the paper, the authors may discuss what are the threats for validity of their proposal. That's what those questions were mostly about, but there are many others that could be addressed. Here are the previuos questions: Is there a limit on the number of connections a web browser is able to accept? What happens when this limit is reached? How about the different Internet connection speed of each browser? Does this affect the browser network overall performance? How does this overhead of connections affect the user speed of operation? How does it affect the user computer?

pp.8, line 238-241:
pp.9, Figure 9:

We modify the figure 9, and we also add the description about the scale of the figure 9 according to the following comment.

The scale of Figure 9 should be further improved in order to better see the actual values of the bars, in in the text referring to it it should be pointed out the scale used and why.

Reviewer 2 Report

The revision addressed my comments. But the narrative is lack of logic and need significant language assistance from native speakers. The sentences in the abstract should be rearranged to become smoothly readable.

Author Response

(The authors gave the same response as above.)
